# Three-dimensional distribution of fine particulate matter concentrations and synchronous meteorological data measured by an unmanned aerial vehicle (UAV) in Yangtze River Delta, China

Si-Jia Lu<sup>1</sup>, Dongsheng Wang<sup>1</sup>, Xiao-Bing Li<sup>1</sup>, Zhanyong Wang<sup>1</sup>, Ya Gao<sup>1</sup>, Zhong-Ren Peng<sup>1,2</sup>

<sup>5</sup> <sup>1</sup> Center for UAV Application and ITS Research, State Key Laboratory of Ocean Engineering, School of Naval Architecture, Ocean & Civil Engineering, Shanghai Jiao Tong University, Shanghai, 200240, China <sup>2</sup> Department of Urban and Regional Planning, University of Florida, Gainesville, FL 32611-5706, USA

Correspondence to: Z. R. Peng (zpeng@dcp.ufl.edu)

- Abstract. Three-dimensional distribution of fine particulate matter ( $PM_{2.5}$ ) and meteorological factors are of great importance to clarify the formation mechanism of haze pollution and to help forecast atmospheric pollution under different meteorological conditions. The objective of this study was to measure  $PM_{2.5}$  concentrations and meteorological data at 300-1000 m altitude using an unmanned aerial vehicle (UAV) equipped with mobile instruments. The study was conducted in a 4 × 4 km<sup>2</sup> space in Lin'an, Yangtze River Delta (YRD), China. The UAV was operated repeatedly for four times in one day along the designed route spirally from the ground to 1000 m altitude with a total of 8 layers and a 100m interval between
- 15 two adjacent layers for five days from 21th August 2014 to 2nd February 2015. PM<sub>2.5</sub>, air temperature, relative humidity, dew point temperature and air pressure were measured during the data collection. The study results indicated that the PM<sub>2.5</sub> concentrations decreased with altitude at 300-1000 m and the variations of PM<sub>2.5</sub> with altitude in morning flights were much bigger than in afternoon flights. Besides, the PM<sub>2.5</sub> concentration levels in morning flights were generally lower than in afternoon flights. PM<sub>2.5</sub> concentrations were positively correlated with dew point temperature and pressure, but positively
- 20 correlated with relative humidity only on pollution days in autumn or winter. The vertical gradient of  $PM_{2.5}$  concentrations was small in pollution days compared with on clean days. These findings provide the key theoretical foundation for  $PM_{2.5}$  pollution forecast and environmental management.

# **1 INTRODUCTION**

Ambient air pollution is a primary environmental problem in industrial and developing countries, and China is not spared
(Ouyang et al., 2013). Among all the air pollutants, the fine particulate matter is responsible for the climate change and visibility degradation (Davidson et al., 2005). What's more, the fine particulate matter pollution threatens the human health, especially cardiorespiratory system (Pope et al., 2002, 2009; Dominici et al., 2006).

Studying the formation and dissipation mechanism of fine particulate matter is of significance for air pollution forecast and urban planning. However, most of the previous studies mainly focused on the characteristics of fine particulate matter on

surface level. The only ground-based observations were not sufficient for further research of the trans-boundary transport of pollutants (Ding et al., 2009), the influence of atmospheric microcirculation (e.g. sea-land breeze, urban heat island effect) on pollutant distribution in urban area (Strawbridge et al., 2004), and calibration of atmospheric model (Boy et al., 2006; Ding et al., 2009).

- Studies of the particulate matter (PM) vertical concentrations in troposphere and planetary boundary layer (PBL) were routinely conducted by meteorological tower (Ding et al., 2005; Yang et al., 2005; Gao et al., 2012; Sun et al., 2013), tethered balloon (Clark et al., 2000; Maletto et al., 2003; McKendry et al., 2004), LiDAR (Strawbridge et al., 2004) and manned aircraft (Wang et al., 2008). Meteorological tower monitoring is fit for long-term continual observations, but it is limited to monitoring elevation (no more than 350 m) and mobility. Ding et al. (2005) and Yang et al. (2005) found that
- PM<sub>2.5</sub> mass concentrations logarithmically decreased with increasing altitude. Ding et al. (2005) also classified the PM<sub>2.5</sub> vertical distribution in Beijing into two patterns: gradual decline pattern and rapid decline pattern. However, Sun et al. (2013) indicated that the reverse distribution of relative humidity (RH) might result into a "higher-top and lower-bottom" pattern of PM<sub>2.5</sub> distribution, considering moisture absorption effect of particulate matter. Tethered balloon monitoring could overcome some drawbacks of meteorological tower monitoring and could obtain PM concentrations at continuous height levels.
- Nevertheless, it is restricted to the horizontal monitoring range. Maletto et al. (2003) indicated that fine particulate matter (PM<sub>2.5</sub>) tended to be well mixed vertically during daytime whereas coarse particulate material (PM<sub>10</sub>) tended to be found close to ground and sources due to gravitational settling. While Mckendry et al. (2004) claimed that layers of fine particulate matter may be found in wintertime nocturnal settlings. LiDAR is a simple and efficiency method to monitoring the PM concentrations in the whole atmosphere indirectly. Strawbridge et al. (2004) discovered that increased PM concentrations in
- the Northeastern valleys during the night flights due to a persistent sea breeze. Compared with the methods discussed above, manned aircraft monitoring is responsible for a large area of monitoring. Wang et al. (2008) found that fine particles are dominant fraction of particulate matter in the troposphere and PM<sub>2.5</sub> concentration distribution is close association with boundary-layer structure and changing of wind.

To better understand and study the dispersion and distribution of PMs, there are several commonly used methods, among

- which unmanned aircraft vehicle monitoring (UAVM) is a state-of-the-art technique. In terms of cost-efficiency, UAVM has an advantage compared with manned aircraft monitoring. Unmanned aircraft monitoring platform (UAMP) were recently applied for monitoring meteorological data (Kroonenberg et al., 2008, Wildmann et al., 2013; Wildmann et al., 2014; Altst ädter et al., 2015), atmosphere structure (Thomas et al., 2012; Wildmann et al. 2014; Lothon et al. 2014) and particulate matter (Clarke et al., 2002; Harnisch et al., 2009; Bates et al., 2013; Altst ädter et al., 2015). Clarke et al. (2002) indicated
- that UAMP is of good quantitative performance under a range of conditions and is able to provide reliable partially dried size distributions. Ramana et al. (2007) found that the simulated clear sky heating rates are consistent with the broadband heating rates observed by stacked aircraft within experimental errors (about 15%). Harnisch et al. (2009) discovered an asymmetry in the aerosol distribution in the cross-valley direction and he presumed that it is related to differences in orientation and albedo of the two valley slopes. Bates et al. (2013) found that frequent aerosol layers aloft with high PNC and enhanced

aerosol light absorption. Altst ädter et al. (2015) discovered that a particle burst event occurred during the boundary layer development in the morning. In summary, most research focused on the vertical distributions of PM concentrations, but few studied the three-dimensional distribution of PM, especially during the formation and dissipation events.

- In our research, a fixed-wing unmanned aircraft vehicle (UAV) was modified from an air-mapping aircraft with the payload 5 capacity of ~5kg. The sensors for PM concentrations and meteorology were calibrated and tested in the laboratory. The UAV research was operated to investigate three-dimensional distribution of PM concentrations both spatially and temporally. 20 monitoring flights in five days across over a half year were totally done over suburban area of Lin'an in YRD. The observations were conducted in the lower troposphere, especially within the atmospheric boundary layer. The daily variety of PM<sub>2.5</sub> in the atmosphere were captured and the accumulation and dissipation of PM<sub>2.5</sub> is discussed in this article. This
- 10 study provides detailed measurements of the small-scale three-dimensional variability of the PM2.5 concentrations.

#### 2 EXPERIMENTS AND METHODS

#### 2.1 Experimental Site

Mobile vertical monitoring were performed in a  $4 \times 4 km^2$  suburban area in the north part of Lin'an, China (E 118 51'-119 52', N 29 56'-30 23') (see Figure 1). The experimental site is approximately 13 km from Lin'an downtown. A highway runs west-to-east across the experimental area. Some machinery manufacture plants are located on the south side of the highway. No direct pollution sources are within or near around the experimental area. There is a relative low density of houses in this area and only hills are surrounded on the southeast and northwest of the area. In the experimental area, nearly half of the ground is covered by trees and one-third of the surface is bare land.

20 Figure 1. Experimental site in Lin'an, China.