# Peer review of "Three-dimensional distribution of fine particulate matter concentrations and synchronous meteorological data measured by an unmanned aerial vehicle (UAV) in Yangtze River Delta, China"

_Atmospheric Measurement Techniques, 2016_

## Referee Comment (RC1) · Anonymous Referee #1 · 29 Mar 2016

***Referee comments*** on "Three-dimensional distribution of fine particulate matter concentrations and synchronous meteorological data measured by an unmanned aerial vehicle (UAV) in Yangtze River Delta, China" by S.-J. Lu et al.

**General comments**

The manuscript describes measurements of fine particulate matter and meteorological parameters with an unmanned aerial vehicle. The aim of these measurements is to characterize the 3D distribution of PM2.5 and meteorological factors, which could be important to clarify the formation mechanisms of haze pollution and to help the forecast of atmospheric pollution. As such, the manuscript represents a substantial contribution to scientific progress within the scope of this journal, since it uses a new, though not completely original, methodology. In fact, some previous works about the application of UAV to the characterization of vertical profiles of atmospheric parameters, but also of particles as measured by optical particle counters (e.g., Brady et al., 2016; Renard et al., 2016) have been already published. These papers should be at least referenced in the work. The scientific approach and applied methods seem generally valid, although some appropriate references are missing. Finally, the number of the figures seems adequate, although the figures of the original version are not of enough quality. Apart from my specific comments below, it is very difficult to follow the presentation of scientific results due to the inappropriate use of English language: apart from some specific comments below, I strongly suggest that the English language of the manuscript is improved through the help of a native English-level speaker, since the language is not fluent and precise. The paper represents a sort of first study of the feasibility of the application of this kind of measurements, and as far as I understood more measurements are planned in the future. As such, conclusions reached are not substantial, and are also related to the main drawbacks of the adopted methodology (e.g., endurance of the batteries and limited payload of the UAVs). The description of experiments and calculations is sufficiently complete and precise to allow their reproduction by fellow scientists (traceability of results). Overall, I believe that a correct rewording of the language of the manuscript, including also the homogenization of tenses, is necessary prior it can be considered for publication on this journal.

**Specific comments**

**Abstract**

Page 1, line 9: Since this sentence is rather general and not restricted to your specific work, you should remove $PM_{2.5}$, since fine particulate matter could be also $PM_1$.

Page 1, lines 21-22: Probably it would be better to rephrase: "These findings are crucial for correct PM2.5 pollution forecast and environmental management."

**Introduction**

Page 1, line 25: Is only fine particulate matter responsible for climate change and visibility degradation? Are there any other important references for aerosols effect on climate?

Page 1, line 27: Dissipation is not the correct term here.

Page 1, lines 27-28, page 2, lines 1-4: It would be necessary to clarify the state of the art of these researches and why studies of vertical profiles are necessary for proceeding in our basic understanding of some mechanisms.

Page 2, lines 5-23: These lines are of key importance for the Introduction and for understanding how fine particulate matter measurements with UAV could help bridging the gap of our present understanding. It is therefore necessary here to correctly understand the pros and cons of each technique capable of retrieving vertical profiles (i.e., meteorological tower, LIDARs, manned aerial vehicles, unmanned aerial vehicles). After that, it is necessary here that the present state of the art about 3D distribution of fine particulate matter is correctly conveyed. I suggest therefore to dive this paragraph in two ones, the first one dealing with drawbacks and advantages of the techniques capable of retrieving vertical profiles of meteorological and atmospheric parameters, and the second one conveying the state of the art on the subject. In this respect, in the revision please mind that generally, it is more convenient to write sentences in this manner:   example --   Ozone is an important greenhouse gas, especially in the upper troposphere [reference 1;  reference 2].   This is much better than:    Reference 1 studied ... and found that ozone is an important greenhouse gas in the UT.  Reference 2 examined .... and concluded that ozone in the UT has a significant greenhouse effect.

Page 2, lines 24-34, page 3, lines 1-3: Similar to my previous comment, please modify the way you refer to literature. This paragraph should explain the advantages, but also drawbacks, of the application of UAV with respect to other techniques. After that, previous findings obtained through the use of UAV should be summarized. Are there differences between UAVs, UAVM and UAMP? If not, please use one term only.

Page 2, line 30: How can UAMP provide partially dried size distributions? This sentence is not clear enough.

Page 3, line 3: Again, dissipation is not the correct term here.

Page 3, lines 4-10: This paragraph comprehends lots of information which should be provided in the Experimental section. You should provide the objective of this work and the structure of the article here.

Page 3, lines 6-7: Do you mean that your objective was to investigate spatial and temporal variations of the 3D distribution of PM concentrations? How can five days cover half a year? Temporal is a quite general term, and 5 days are not sufficient to achieve a correct and exhaustive characterization.

Page 3, lines 13-14: It is rather uncommon to find coordinates expressed in this order (i.e., E/W and S/N before numbers, and longitude before latitude.

Page 3, lines 15-16: Please quantify the term "some" and the kind of manufacture, this could be important for the characterization of particulate matter.

Page 3, lines 16-17: Please quantify what relative low density mean.

Page 3, lines 13-18: This paragraph could benefit from the addition of climatological information about the experimental site or at least for Lin'an, to better characterize it. Moreover, references for the information should also be provided.

Page 4, lines 1-7: It would be necessary to provide more technical details about the UAV (weight, maximum payload, …).

Page 4, line 6: It would be necessary to provide the speed in m/s, so that it could be really straightforward to understand how long the UAV stood in each layer. Please indicate here also the spatial interval between each vertical layer.

Page 4, line 6: Without knowing the time resolution of the measurements it is not possible to understand if 45 minutes are really enough. It would be better to cite the table where the time resolution of your measurements is provided here.

Page 4, line 14: Black carbon is not used in this work so it is quite confusing that you put it here. However, it would be interesting to know how black carbon is related to the other measurements. The authors should provide the sensor used by the aethalometer to provide pressure measurements.

Page 4, lines 11-16 and Table 1: From Table 1, it is apparent that the instruments onboard the UAV have different time resolutions. How did you make your data homogenous? Moreover, in case of such high time resolution, some sensors such as the aethalometer can be affected by noise problems. How did you make sure that noise did not affect your measurements? Finally, further problems may also arise due to vibrations, turbulence, electrical interferences: more technical details should be provided here.

Page 5, line 4: The correlation coefficient alone is not enough to take for granted the consistence between particle mass concentration instruments and TEOM mass measurements:  the intercept should be also provided here.

Page 5, lines 7-8: "assigned" is not the correct term here.

Page 5, line 8: The information for the duration of the flight is different from the one reported on Page 4, line 7. Please be consistent.

Page 6, line 5: It is not clear what "checked to zero" means in this context. Please provide a better explanation.

Page 6, lines 6-8: This information should be provided before when describing the instruments onboard the UAV.

Page 6, lines 9-11: Explain a little bit the characteristics of the meteorological station (e.g., location, altitude, …)

Page 6, lines 11-14: You should retrieve and discuss the planetary boundary layer height, a parameter which could be very important for your investigations. More importantly, how can you relate measurements taken onboard the UAV with a 1-2 s time resolution taken at different hours of the day with sounding measurements taken every day at the same hour?

Page 6, lines 21 and 22: "self-monitoring" is not the correct term here.

Page 7, lines 1-2: As far as I understood, the TSI SIDEPACK AEROSOL is not an OPC as you report here, but it is rather an impactor, so it is not based on the light scattering principle.

Page 7, lines 7-8: It would be more convenient to use a coarser time resolution due to problems of noise for some instruments, and equal for all instruments onboard.

Page 7, lines 11-13, and Page 8, Figure 3: In the Experimental section (Page 7, lines 7-8), it seemed that an average for each height layer was retrieved losing the latitude-longitude variation, which is not the case as discussed in the sentences and as shown in the Figure 3. Please explain better that the average vertical variability was also retrieved, but also 10 s averages were separately analysed in order to examine the latitude-longitude-height (therefore 3D as reported from the Title) distribution. After that, since the vertical variation is more pronounced than the horizontal variability, it is correct to average each vertical layer to better examine the vertical $PM_{2.5}$ profiles.

Page 9, lines 7-20: These results could be discussed much better if you retrieved also the PBL height as I previously suggested. The English language should be greatly improved here.

Page 9, line 8: Probably you mean that the low PBL height limits the vertical distribution of $PM_{2.5}$, but this sentence is not so clear.

Page 9, lines 13-14: A range such that you report cannot be preceded by about.

Page 9, line 17-20: Please check the number of significant digits.

Page 9, lines 16-20: Rephrase, these sentences are not clear enough.

Page 10, Table 3: Check the units for the vertical gradient.

Page 10, Figure 5: You should briefly remind here that wind speed and direction measurements were taken

Page 11, lines 4-17, page 12, lines 1-2 and 6-13, page 13, lines 4-10: These sentences need substantial rewording. How can you relate sounding measurements take once a day with your vertical measurements averaged to 10 s?

Page 11, line 17: Check number of significant digits.

Page 12, lines 12-13: Increasing wind speed with increasing altitude could have some basic explanations, which you could try to provide here.

Page 13, lines 12-16: These sentences need substantial rewording. It is not clear how the consistency between sounding and UAV data indicates the feasibility and utility of UAV measurements.

Page 15, lines 2-4: Pearson's correlation coefficient is probably not appropriate here, since strictly speaking Pearson's correlation can be applied only in the case of normal distributions, which probably is not the case here (however, if it is the case you can just briefly discuss that).

Page 16, line 2: The only positive correlation which is apparent from the table is the one of PM2.5 with dew point. The correlations with pressure and temperature are instead observed only in some situations, and are therefore probably not conclusive.

Page 16, lines 2-6: The correlation of PM2.5 with relative humidity can be explained by some physical mechanisms. Please try to explain.

Page 16, lines 8-22: Conclusions should be greatly revised both for the language as well as after the appropriate revisions concerning other sections have been undertaken.

**Technical comments**

Page 1, line 13: Delete "for". Change "in one" with "a".

Page 1, line 14: Delete "the".

Page 1, line 16: Change "during the data collection" with "along the flight".

Page 1, line 18: Change "bigger" with "larger".

Page 1, line 21: Change "in" with "on", delete "on".

Page 1, line 28: Change "of significance" to "significant".

Page 1, line 29: Change the second "on" to "at".

Page 2, line 1: Delete "The only". Change "were" to "are". Change "further research" to "a correct understanding".

Page 2, line 3: Change "model" to "models".

Page 2, line 5: Delete "the". Change "concentrations" with "profiles".

Page 2, line 8: Change "continual" with "continuous".

Page 2, line 9: Change "to monitoring" to "in terms".

Page 2, line 18: Change "efficiency" to "efficient".

Page 2, line 21: "responsible" is not the correct term here.

Page 2, line 30: Change "Is of good quantitative performance" with "has a good performance".

Page 3, line 4: Change "a" with "the".

Page 3, lines 6-7: Rephrase: "A total of 20 monitoring flights over half a year were carried out over the suburban area of Lin.an in YRD."

Page 3, lines 8-9: Rephrase: "The diurnal variation of PM2.5 as well as its accumulation and dissipation in the atmosphere were captured and are discussed in the present article."

Page 3, line 10: Delete the second "the".

Page 3, line 13: Change "were" to "was".

Page 3, line 13: Change "were" to "was".

Page 3, line 14: Add "distant" before "from".

Page 3, line 16: Change "near around" to "close".

Page 4, line 5: Change "its" to "a" and "lands" to "landed".

Page 4, line 6: Add a space before "300".

Page 4, line 6: Add "of about" before "120" and delete ~.

Page 5, line 5: Change "on a total" with "for".

Page 5, line 6: Delete "including 16 flights". Rephrase the second sentence: "Four flights …, for a total of 16 flights (Table 2)."

Page 6, line 2: Move "such as remaining battery and storage space" before "were".

Page 6, lines 2-3: Rephrase: "… and a visual inspection was conducted to determine the eventual compression of inlet tubing at curve."

Page 6, line 4: Delete "allowed to" and change the tense of "warm up".

Page 6, line 11: Rephrase: "…), maintained by the Meteorological Bureau of Lin'an."

Page 6, lines 11-16: Rephrase: "Sounding meteorological data (air temperature, dew point temperature, relative humidity, wind speed and wind direction) from the sounding station located in Hangzhou, China, located about 40 km away far away from the experimental site which operates soundings at 12:00 UTC every day were downloaded from the University of Wyoming (…)."

Page 6, line 22: Change "when the UAV was taking off" to "during take-off".

Page 8, Figure 3: The Figure has a bad resolution. Moreover, scale units for the legend should be provided.

Page 9, line 2: "fligh-3" should be "flight-3". Delete the second comma. Add "as" before "afternoon".

Page 9, line 3: Move "increasing" before "altitude".

Page 9, line 4: Change "depicts" to "correspond to". Unify the two sentences as: "…, 2013) consistently with results from tower observations (…)".

Page 9, line 7: Delete "in the".

Page 9, line 13: Change "more" to "higher".

Page 9, line 16: Change "Ding (Ding et al., 2005)" to "Ding et al. (2005)".

Page 13, line 4: Change "particle" to "particulate".

**References**

Brady J.M., Stokes M.D., Bonnardel J., Bertram T.H., 2016. Characterization of a quadrotor unmanned aircraft system for aerosol-particle-concentration measurements. Environmental Science and Technology 50, 1376-1383, doi:10.1021/acs.est.5b05320

Renard J.-B., Dulac F., Berthet G., Lurton T., Vignelles D., et al., 2016. LOAC: a small aerosol optical counter/sizer for ground-based and balloon measurements of the size distribution and nature of atmospheric particles – Part 2: First results from balloon and unmanned aerial vehicle flights. Atmospheric Measurement Techniques Discussion 8, 1261-1299.

---

## Referee Comment (RC2) · Anonymous Referee #2 · 13 Apr 2016

General The paper presents a setup and results of measurements of PM2.5 and BC concentrations and basic meteorological parameters by using an unmanned aerial vehicle (UAV) at a site in eastern China. With this setup the 3-dimensional distribution of pollutants can be measured which is undoubtedly very useful, for instance for evaluating the performance of air pollution forecasting models. In principle the work is ok and worth publishing but there are some shortcomings.

I am not rejecting this paper but I am not quite sure that this journal, AMT, would be

the correct one for this work. My opinion is based on that as far as measurement methods, data processing , or algorithms or such are concerned, there are actually no new technical or methodological innovations or observations in this paper. Numerous aerosol measurements using UAVs have been published during the last few decades. So, as a concept there is nothing new. The instruments installed in the aircraft are all commercial, made by well known manufacturers. In the data processing there are essentially no new innovations either. However, as I wrote above, the results of this work are useful, for instance for evaluating the performance of air pollution forecasting models. Therefore a more suitable journal would be one that has the atmosphere itself as its focus, not methods as in AMT.

Detailed comments

P1,L25: "... Among all the air pollutants, the fine particulate matter is responsible for the climate change and..." This is not true. Particles do have radiative effects for sure, but mainly cooling. It is the greenhouse gases that are responsible for climate change. If all aerosol emissions were stopped, climate change would continue due to GHGs, probably faster.

The language should be corrected. As a non-native English speaker I don't want to be too picky, my English is not error free either. But in this text there are very many sentences and expressions that make it difficult to read and need correction. Here are just a few examples:

P1,L28: I would recommend changing the word "dissipation" to dispersion. Actually, the word "dissipation" is used in several sentences of the paper but in none of them it is really the good term.

P2,L1: " The only ground-based observations were not sufficient..." Rewrite as "Ground-based observations only are not sufficient..."

P2,L5-6 : " Studies ... were routinely conducted by meteorological tower..." This means

that there was a meteorological tower who conducted studies. In other words, the expression "were conducted by" gives an impression of that the tower is a person. And the tense is misleading: "were conducted" suggests that they are not conducted any more.

P2,L8-9: " .. it is limited to monitoring elevation (no more than 350 m) and mobility." This would mean something is monitoring elevation, in other words monitoring whether someting is rising or falling. And "monitoring mobility" would mean monitoring, whether something is moving. Rewrite.

A related note: the word "monitoring" is used throughout the text in an uncorrect way. In aerosol science monitoring generally means long-lasting, continuous measurements. For example at an air quality measurement station. The UAV measurements in this paper are not monitoring, unless the flights are more or less continuous, which they are not.

P2, L25: " unmanned aircraft vehicle" Should be "unmanned aerial vehicle"

Section 2.2

What is the manufacturer and model of the aircraft? What is its fuel.

Table 1. The aethalometer does not measure pressure. What did you use for measuring p?

P5.L2-4: Describe the results of the intercomparison better than giving just one correlation coefficient. How long was the intercomparison, show scatter plots, and regression lines, slope and offset..

Table 2 needs a proper, detailed caption and column explanations.

P6,L21 What does "self-monitoring" mean?

P7,L5. The formula (1) is not from Day et al. (2000), where is it from? They gave growth of scattering coefficient and it is not the same as the growth of mass. And it

varies with the chemical composition. –what is the reasoning for using this CF? An it is unclear from the explanation, how did you use the CF. Did you correct with it all PM concentrations to RH 0% or what? Secondly, if I put in the formula any RH > 2 and 100, the formula gives negative values. Correct it.

In the time series figures, add dates in the x-axes to make it easier to read.

Move section 3.4 earlier because the met data are used in the explanations of the profiles.

Table 5 is useless.

---

## Author Comment (AC1) · 1 Jun 2016

Manuscript Ref: Manuscript ID AMT-2016-57

Original Title: Three-dimensional distribution of fine particulate matter concentrations and synchronous meteorological data measured by an unmanned aerial vehicle (UAV) in Yangtze River Delta, China

Modified Title: Three-dimensional distribution of fine particulate matter concentrations

and synchronous meteorological data measured by an unmanned aerial vehicle (UAV)

Dear Editor and All Reviewers,

Thank you for your suggestions and comments on the manuscript AMT-2016-57. We have completed the revision of our manuscript which can be found in the attachment.

We consider your opinions and suggestions as great improvements to our manuscript, and have meticulously addressed each of your comments to meet the requirements of its publication in Atmosphere Measurement Techniques.

The main revisions are listed below:

1.Removing grammar mistakes

2.Rewrote most parts of the paper to make it more rigorous in expression and more compact around the theme of the paper.

3.Refined the experiment design, data collection, and data processing to verify the transferability of the study to other settings.

4.Revised the methodology to make it clearer for implementation.

5.Added more explanations on measurement and modeling results to make the study easier to understand.

6.Updated figures and tables with more concise annotations and discussions.

7.Refined the discussion throughout the paper with reasonable explanations of findings.

8.Rephrased the ambiguous words and sentences in a more intuitive meaning.

9.Polished the whole original manuscript.

Attached please find our detailed responses to three referees' comments and suggestions, in which we answered their criticisms and comments item by item. We would

like to thank you again for your valuable comments and suggestions in improving our manuscript. We look forward to your response.

Sincerely,

Si-Jia Lu, Dongsheng Wang, Xiao-bing Li, Zhanyong Wang, Ya Gao, and Zhong-Ren Peng

Reply to the First Referee – AMT-2016-57

Please refer to the manuscript in PDF format about line and page numbers.

Reviewer #1:

General Comments

1. Some previous works about the application of UAV to the characterization of vertical profiles of atmospheric parameters, but also of particles as measured by optical particle counters (e.g., Brady et al., 2016; Renard et al., 2016) have been already published. These papers should be at least referenced in the work.

Reply: Thanks for the referee's comments. It certainly has some previous works about the application of UAVs and of OPCs, The related studies have been supplemented as references in the revision, including your mentioned ones [see P1, L33; P2, L14; P3, L17].

2. The scientific approach and applied methods seem generally valid, although some appropriate references are missing.

Reply: Thanks to the referee for pointing out this problems. Some studies with UAV field experiments have been supplemented as references in the updated manuscript [see P3, L17; P13, L3].

3. Finally, the number of the figures seems adequate, although the figures of the original version are not of enough quality.

[Figure]

Reply: Sorry about the bad quality of the figures. All figures in the manuscript have been checked one by one and are updated with good quality ones if necessary [see Fig. 2 on Page 4, Fig. 3 on Page 8, Fig. 4 on Page 10, Fig. 6 on Page 13, Fig. 8 on Page 15, Fig. 9 on Page 20].

4. Apart from my specific comments below, it is very difficult to follow the presentation of scientific results due to the inappropriate use of English language: apart from some specific comments below, I strongly suggest that the English language of the manuscript is improved through the help of a native English-level speaker, since the language is not fluent and precise.

Reply: The manuscript has been rewritten and the language throughout the whole paper has been polished to our best. The passages which were ambiguous or difficult to read in our original manuscript have been rewritten. We hope our study can be clearly presented in this revised manuscript. Additionally, we have revised the grammar mistakes with aid of a professor in University of Florida, U.S.

5. The paper represents a sort of first study of the feasibility of the application of this kind of measurements, and as far as I understood more measurements are planned in the future.

Reply: We have certainly conducted several field experiments about fine particulate matter, black carbon and ozone, both in Shanghai and Lin'an, considering the previous experiment experience.

6. As such, conclusions reached are not substantial, and are also related to the main drawbacks of the adopted methodology (e.g., endurance of the batteries and limited payload of the UAVs).

Reply: Thanks for the referee's comment. We have reconstructed and rephrased the conclusions concerning other sections have been revised.

7. Overall, I believe that a correct rewording of the language of the manuscript, including also the homogenization of tenses, is necessary prior it can be considered for publication on this journal.

Reply: Thanks for your suggestions. We have rewritten parts of the content in the revised manuscript to remove these mistakes. Some statements and graphs have been revised based on the referee's suggestions. For detailed information, please refer to the replies to specific comments.

Specific Comments

8. Page 1, line 9: Since this sentence is rather general and not restricted to your specific work, you should remove PM2.5, since fine particulate matter could be also PM1.

Reply: Thanks for the referee's suggestion. We have remove "PM2.5" in the revised manuscript to make the statement more precise [see Page 1, Lines 25].

9. Page 1, lines 21-22: Probably it would be better to rephrase: "These findings are crucial for correct PM2.5 pollution forecast and environmental management."

Reply: Thanks for the referee's suggestion. Your statement is very precise, and this sentence has been revised based on the suggestion [see Page 1, Lines 22].

10. Page 1, line 25: Is only fine particulate matter responsible for climate change and visibility degradation? Are there any other important references for aerosols effect on climate?

Reply: Thank the referee for pointing out these questions. Besides fine particulate matter, the GHGs, like carbon dioxide, are also responsible for the climate change. Several studies have pointed out that aerosol particles in the atmosphere influence clouds, and there by climate [Booth er al., 2012; Stevens et al., 2012; et al.]. These reference citations are added into the vision [see Page 1, Lines 25-29].

11. Page 1, line 27: Dissipation is not the correct term here.

[Figure]

Reply: Thanks for the referee's comment. According to another referee's suggestion, the "dissipation" is changed to "dispersion" [see Page 1, Lines 2; Page 3, Lines 6; Page 3, Lines 11; Page 15, Lines 1; Page 15, Lines 6; Page 15, lines 14; Page 16, Lines 5; Page 16, Lines 18].

12. Page 1, lines 27-28, page 2, lines 1-4: It would be necessary to clarify the state of the art of these researches and why studies of vertical profiles are necessary for proceeding in our basic understanding of some mechanisms.

Reply: Thanks to the referee for giving the constructive comment. Based on literature reviewed, we have supplemented a substantial explanation on the state of the art of these researches and on the necessity of vertical profiles for understanding of some mechanisms in the revision [see Page 2, Lines 1-9].

13.Page 2, lines 5-23: These lines are of key importance for the Introduction and for understanding how fine particulate matter measurements with UAV could help bridging the gap of our present understanding. It is therefore necessary here to correctly understand the pros and cons of each technique capable of retrieving vertical profiles (i.e., meteorological tower, LIDARs, manned aerial vehicles, unmanned aerial vehicles). After that, it is necessary here that the present state of the art about 3D distribution of fine particulate matter is correctly conveyed. I suggest therefore to dive this paragraph in two ones, the first one dealing with drawbacks and advantages of the techniques capable of retrieving vertical profiles of meteorological and atmospheric parameters, and the second one conveying the state of the art on the subject. In this respect, in the revision please mind that generally, it is more convenient to write sentences in this manner: example – Ozone is an important greenhouse gas, especially in the upper troposphere [reference 1; reference 2]. This is much better than: Reference 1 studied ... and found that ozone is an important greenhouse gas in the UT. Reference 2 examined .... and concluded that ozone in the UT has a significant greenhouse effect.

Reply: Thanks to the referee for giving the constructive comments. We have recon-

structed the Introduction according to the referee's suggestion. The pros and cons of each technique are stated clearly in the paragraph and the state of the art on the subject is illustrated in the next paragraph in the updated manuscript. The sentences have been rewritten based on the referee's advice [see Page 2, Lines 10-26].

14. Page 2, lines 24-34, page 3, lines 1-3: Similar to my previous comment, please modify the way you refer to literature. This paragraph should explain the advantages, but also drawbacks, of the application of UAV with respect to other techniques. After that, previous findings obtained through the use of UAV should be summarized. Are there differences between UAVs, UAVM and UAMP? If not, please use one term only.

Reply: Thanks for the referee's suggestions and comments. The references in the while manuscript have been modified according to the referee's suggestion. We have supplemented a substantial explanation on the pros and cons of the UAV techniques. Previous findings obtained with UAV technology are summarized in the updated manuscript [see Page 2, Lines 27-34 & Page 3, Lines 1-6]. Sorry for the confusion. The terms have been unified with UAVs [see Page 2, Lines 28, Lines 30].

15. Page 2, line 30: How can UAMP provide partially dried size distributions? This sentence is not clear enough.

Reply: Sorry for the confusion. The UAMP developed by Clarke (2002) consists of a customized mini OPC and a RH/temperature sensor. It can provide 256 size bins over the 0.3–14-$\mu$m diameter range and relative humidity. The measured particle concentrations were calibrated with RH, considering particle size changes associated with humidity changes. The sentence has been rephrased in the revision [see Page 2, Lines 33].

16. Page 3, line 3: Again, dissipation is not the correct term here.

Reply: Thanks for the referee's comment. The "dissipation" is changed to "dispersion", according to another referee's suggestion [see Page 3, Lines 6].

[Figure]

17. Page 3, lines 4-10: This paragraph comprehends lots of information which should be provided in the Experimental section. You should provide the objective of this work and the structure of the article here.

Reply: Thanks to the referee for giving the constructive comments. The paragraph has been reconstructed in the revision according to the referee's suggestion. [see Page 3, Lines 7-12].

18. Page 3, lines 6-7: Do you mean that your objective was to investigate spatial and temporal variations of the 3D distribution of PM concentrations? How can five days cover half a year? Temporal is a quite general term, and 5 days are not sufficient to achieve a correct and exhaustive characterization.

Reply: Sorry for the confusion. In this study, one major objective was to investigate the 3D distribution of PM concentrations and to explore the relationship of diurnal distribution variations. It has been rephrased in the revision [see Page 3, Lines 7-10].

We certainly agree with the referee's opinion. Considering the personnel arrangements and equipment failures, we just conducted experiments for 5 days from August 2014 to February 2015. However, we have implemented more field experiments later both in Lin'an and Shanghai. And what's more, joint experiment with tethered balloon was also carried out in Fengxian, Shanghai. We think these work are essential for further studying PM pollution mechanism and its impact on the environment. It has been revised in the updated manuscript to make it more clear.

19. Page 3, lines 13-14: It is rather uncommon to find coordinates expressed in this order (i.e., E/W and S/N before numbers, and longitude before latitude.

Reply: Thanks for the referee's comment. We have corrected them [see Page 3, Lines 15-16].

20. Page 3, lines 15-16: Please quantify the term "some" and the kind of manufacture, this could be important for the characterization of particulate matter.

Reply: Thanks for the referee's suggestions. The manufactures have been specified and further explanations about the impact of these manufactures on local atmosphere have been implemented in the updated manuscript [see Page 3, Lines 20].

21. Page 3, lines 16-17: Please quantify what relative low density mean.

Reply: Thanks for the referee's comment. The exact population in the experimental site is hard to get, but residential land is less than 10% of the experimental area. It has been refined [see Page 3, Lines 22-24].

22. Page 3, lines 13-18: This paragraph could benefit from the addition of climatological information about the experimental site or at least for Lin'an, to better characterize it. Moreover, references for the information should also be provided.

Reply: Thanks for the referee's constructive comments. Lin'an has a subtropical monsoon type climate with four quite distinct seasons and north eastern winds prevail in winter, whilst southerly winds reign in summer. The experimental site is approximately 13 km distant from the downtown. The Changxi Provincial Highway runs alongside the Zhongtiaoxi River west-to-east across the experiment site. A hi-tech development zone located on the southern side of the highway is under construction. Thirteen machinery manufacture plants are operated in the hi-tech zone. Some residential areas are located on the northern side of the highway. No direct pollution sources are within or close to the experiment site. Less than 10% of the experiment site is residential land surrounded only by hills on the southeast and northwest of the area. In the experiment site, nearly half of the ground is covered by trees and one-third of the surface is bare land.

We have supplemented the climatological information about Lin'an in new manuscript [see Page 3, Lines 16-24].

23. Page 4, lines 1-7: It would be necessary to provide more technical details about the UAV (weight, maximum payload, . . .).

Reply: Thanks for the referee's comments. It has a wingspan of 2.4 m, a maximum take-off weight of about 15 kg and a maximum payload of 5 kg. The UAV is powered by a gasoline engine with a maximum output power of 2800 kW. Safe operation is possible at a wind speed of 12 m s-1. The cruising speed is typically 30 to 35 m s-1. 2*1.5 L tanks allow flight endurance for over one hour.

More technical information about the UAV, including the wingspan, maximum takeoff weight, maximum payload, and engine power, has been provided in the revision [see Page 4, Lines 8-13].

24. Page 4, line 6: It would be necessary to provide the speed in m/s, so that it could be really straightforward to understand how long the UAV stood in each layer. Please indicate here also the spatial interval between each vertical layer. Reply: Thanks for the referee's suggestions. The unit has been updated according to the referee's advice. The spatial interval between each vertical layer is 50 meters and this has been complemented [see Page 4, Lines 10-14].

25. Page 4, line 6: Without knowing the time resolution of the measurements it is not possible to understand if 45 minutes are really enough. It would be better to cite the table where the time resolution of your measurements is provided here.

Reply: Thanks for the referee's suggestion. Table 1 has been cited in new manuscript [see Page 4, Lines 15].

26. Page 4, line 14: Black carbon is not used in this work so it is quite confusing that you put it here. However, it would be interesting to know how black carbon is related to the other measurements. The authors should provide the sensor used by the aethalometer to provide pressure measurements.

Reply: Thanks for the referee's suggestions and for pointing out the mistake. In our experiments, BC sensor (Aethlabs AE51) and Ozone sensor (POM Ozone Monitor, 2B Technologies, Inc.) were actually on board besides PM sensor. The pressure was

measured by the pressure sensor built-in POM. However, the type of the pressure sensor is not provided in the operation manual, and we have consulted the manufacturer by email but haven't received the response. The mistake has been corrected in the revision [see Table 1 on Page 5]. The referee's suggestion about investigating the relationship of BC with other measurements is an important and meaningful research. Our team is doing research. The relevant papers are under written.

27. Page 4, lines 11-16 and Table 1: From Table 1, it is apparent that the instruments onboard the UAV have different time resolutions. How did you make your data homogenous? Moreover, in case of such high time resolution, some sensors such as the aethalometer can be affected by noise problems. How did you make sure that noise did not affect your measurements? Finally, further problems may also arise due to vibrations, turbulence, electrical interferences: more technical details should be provided here.

Reply: This is a good question. In our experiment, in order to obtain high resolution measurements, the minimum reporting interval of each instrument was chosen within the recommended reporting interval range, i.e., PM sensor for 2 seconds, Temperature/RH sensor for 2 seconds, and Ozone sensor for 10 seconds. These sensors were fixed with antivibration pad. Also, the inlets of the instrument were positioned on the underside of the aircraft to avoid the turbulence effect [see Page 5, Lines 13]. Before experiments, the build-in recording time of all instruments onboard were strictly synchronized with BST (Beijing Standard Time). The outliers of dataset were deleted [see Page 5, Lines 13] and then averaged to 10 seconds in order to eliminate noise [see Page 7, Lines 21]. We have emphasized these details in the revision.

28. Page 5, line 4: The correlation coefficient alone is not enough to take for granted the consistence between particle mass concentration instruments and TEOM mass measurements: the intercept should be also provided here.

Reply: Thanks for the referee's suggestion. The comparison experiment lasted for 21

days and was conducted under different temperature and humidity conditions. The AM510 data is quite consistent with the TEOM data, with correlation coefficient (R) of 0.99, intercept of -11.52 and slope of 0.82 [see Page 7, Lines 10-11]. The picture blow shows the comparison between AM510 and TEOM data on 4 days.

29. Page 5, lines 7-8: "assigned" is not the correct term here.

Reply: Thanks for the referee's comments. To make a clear expression, we have rephrased the related sentences in new manuscript [see Page 6, Lines 7-8].

30. Page 5, line 8: The information for the duration of the flight is different from the one reported on Page 4, line 7. Please be consistent.

Reply: Thanks for pointing out the mistake and we have corrected it [see Page 6, Lines 9].

31. Page 6, line 5: It is not clear what "checked to zero" means in this context. Please provide a better explanation.

Reply: Sorry about the confusion. The explanation of equipment operation has been supplemented and rephrased [see Page 7, Lines 6-7].

32. Page 6, lines 6-8: This information should be provided before when describing the instruments onboard the UAV.

Reply: Thanks for the referee's suggestion. The information about the inlets arrangement has been presented in the [see Page 5, Paragraph 2].

33. Page 6, lines 9-11: Explain a little bit the characteristics of the meteorological station (e.g., location, altitude, . . .)

Reply: Thanks. More information about the meteorological has been supplemented in the new manuscript.

34. Page 6, lines 11-14: You should retrieve and discuss the planetary boundary

layer height, a parameter which could be very important for your investigations. More importantly, how can you relate measurements taken onboard the UAV with a 1-2 s time resolution taken at different hours of the day with sounding measurements taken every day at the same hour?

Reply: This is a good question. The planetary boundary layer height is an undoubtedly important factor for the distribution of PM2.5 concentrations. We had tried to find a commercial PBL instrument that could equipped onboard but failed. We are now seeking cooperate with Lin'an Meteorological Bureau to obtain the PBL height data.

35. Page 6, lines 21 and 22: "self-monitoring" is not the correct term here.

Reply: Sorry for the confusion. What we want to express in the manuscript is that the sampling data, especially PM data, could be contaminated by the exhaust from the UAV engine. We have rephrased the sentences in the revision to avoid misunderstanding [see Page 6, Lines 11-12].

36. Page 7, lines 1-2: As far as I understood, the TSI SIDEPACK AEROSOL is not an OPC as you report here, but it is rather an impactor, so it is not based on the light scattering principle.

Reply: Thanks for the referee's comment. We have double checked the principle of the AM 510 (TSI, Inc.) The build-in PM sensor was a 90° light scattering, 670 nm laser diode. (available at: http://www.tsi.com/SIDEPAK-Personal-Aerosol-Monitor-AM510/). The humidity impact on the sampling PM concentration were discussed in the some papers and notes (see below). It seems that humidity impact is related to the constitution of the particulate matter.

1)Dusttrak DRX Aerosol Monitor in Environmental Application Note EXPMN-066

2)Laulainen, N.S. (1993). "Summary of conclusions and recommendations from a visibility science workshop", Pacific Northwest Lab., Richland, WA (United States), URL: http://www.osti.gov/bridge/servlets/purl/10149541-uEhPL2/.

[Figure]

3)Ramachandran, G., J.L. Adgate, G.C. Pratt, and K. Sexton, Characterizing Indoor and Outdoor 15 Minute Average PM 2.5 Concentrations in Urban Neighborhoods. Aerosol Science and Technology, 2003, 37(1): 33 - 45, doi: 10.1080/02786820300889.

37. Page 7, lines 7-8: It would be more convenient to use a coarser time resolution due to problems of noise for some instruments, and equal for all instruments onboard.

Reply: Thanks for the referee's suggestions. In this study, the short logging interval was chosen to obtain as high as possible resolution of data, considering the aircraft cruising speed up to 30 m s-1. The high resolution data is very helpful for mapping the 3D PM concentration distribution map. As to the noise problem, the instruments were wrapped with foam buffer to avoid the vibration problem [see Page 7, Lines 13]. Also, the PM and Temp/RH data were averaged to 10-second. It seems a lot troublesome, and we will consider seriously optimizing the data logging interval settings in the future.

38. Page 7, lines 11-13, and Page 8, Figure 3: In the Experimental section (Page 7, lines 7-8), it seemed that an average for each height layer was retrieved losing the latitude-longitude variation, which is not the case as discussed in the sentences and as shown in the Figure 3. Please explain better that the average vertical variability was also retrieved, but also 10 s averages were separately analyzed in order to examine the latitude-longitude-height (therefore 3D as reported from the Title) distribution. After that, since the vertical variation is more pronounced than the horizontal variability, it is correct to average each vertical layer to better examine the vertical PM2.5 profiles.

Reply: Thanks for the referee's constructive suggestion. It turns out that the horizontal distribution of PM concentration at each height level is much more homogeneous than vertical distribution. Therefore, we ignored the latitude-longtitude variation and averaged the data for each height layer. We have implemented the corresponding explanation about both vertical and horizontal distribution of PM2.5 concentration in the updated manuscript [see Page 7, Lines 25-29].

39. Page 9, lines 7-20: These results could be discussed much better if you retrieved

also the PBL height as I previously suggested. The English language should be greatly improved here.

Reply: Thanks for the referee's suggestions. The PBL height hasn't be obtained due to a limitation of available PBL instrument. However, we are seeking the cooperation with Lin'an Meteorological Bureau. The manuscript has been rewritten and the language throughout the whole paper has been polished to our best.

40. Page 9, line 8: Probably you mean that the low PBL height limits the vertical distribution of PM2.5, but this sentence is not so clear.

Reply: We fully agree with the referee's opinion and the corresponding sentence has been rewritten in the revision [see Page 8, lines 6-10].

41. Page 9, lines 13-14: A range such that you report cannot be preceded by about. Reply: Thanks for the referee's suggestion. This mistake has been corrected in the new manuscript [see Page 9, lines 12].

42. Page 9, line 17-20: Please check the number of significant digits.

Reply: Thanks for the referee's suggestion. The number of significant digits have been unified in the revised manuscript [see Page 9, lines 15-17].

43. Page 9, lines 16-20: Rephrase, these sentences are not clear enough.

Reply: Thanks for the referee's comment and the corresponding sentences has been rephrased in new manuscript [see Page 9, lines 14-17].

44. Page 10, Table 3: Check the units for the vertical gradient.

Reply: Thanks for the referee's suggestion. We think "the Slope of the vertical profiles" is more suitable to express our opinion compared with "the vertical gradient". Therefore, it has been corrected in the revision [see Table 3 on Page 9].

45. Page 10, Figure 5: You should briefly remind here that wind speed and direction

measurements were taken.

Reply: Thanks for the referee's suggestion. The data source and the meteorology station information have been implemented in new manuscript [see Fig. 6 on Page 13, Fig. 8 on Page 14 & Fig. 9 on Page 15].

46. Page 11, lines 4-17, page 12, lines 1-2 and 6-13, page 13, lines 4-10: These sentences need substantial rewording. How can you relate sounding measurements take once a day with your vertical measurements averaged to 10 s?

Reply: Thanks for the referee's constructive suggestions. The sounding are operated twice a day (at UTC 0:00 and 12:00 everyday). Three groups of sounding wind speed and direction were plotted in Fig. 7.ïïjĹThe first group was sampled 12 hours before the flight 1. The second group was sampled during the flight 1. And the third group was sampled 12 hours after the flight 1ïïjĽ. It is certainly sure that sounding measurements taken once a day (exactly every 12 hours) are not strong enough to interpret the diurnal variations of the high resolution PM data. But, we hope it can present us an evolution trend of the wind filed at the 300-1000 m height and hence provide us the evidence of the pollution formation and dispersion. We have implemented more detailed explanation of the sounding data and its possible relationship with the sampled PM concentrations in the updated manuscript [see Page 14, lines 5-19, Page 15, lines 7-14, Page 16, lines 5- 12].

47. Page 11, line 17: Check number of significant digits.

Reply: Thanks for the referee's suggestion. The number of significant digits have been unified in the revised manuscript [see Page 14, lines 15-19].

48. Page 12, lines 12-13: Increasing wind speed with increasing altitude could have some basic explanations, which you could try to provide here.

Reply: Thanks for the referee's suggestion. The wind speed with increasing altitude could be the reduced air friction in higher atmosphere. The explanation has been
added into the revision, together with reference [see Page 15, lines 12].

49. Page 13, lines 12-16: These sentences need substantial rewording. It is not clear how the consistency between sounding and UAV data indicates the feasibility and utility of UAV measurements.

Reply: Thanks for the referee's comments. Some explanations in the original manuscript are indeed a bit ambiguous. The sounding campaign and the flight-1 measurement are generally synchronous. Therefore, the general trend of flight-1 data were consistent with the sounding data since they are comparable. This could show the feasibility and utility of UAV in atmosphere research in some extent. Additionally, the UAV sampling could provide higher resolution than sounding within the PBL and lower troposphere. To avoid possible confusion and misunderstanding, these explanations mentioned above have been implemented in the revised manuscript [see Page 16, lines 15].

50. Page 15, lines 2-4: Pearson's correlation coefficient is probably not appropriate here, since strictly speaking Pearson's correlation can be applied only in the case of normal distributions, which probably is not the case here (however, if it is the case you can just briefly discuss that).

Reply: Thanks for the referee's suggestion. In our previous work (Wang et al., 2015), the normality test between PM2.5 concentrations and meteorological parameters has been conducted in our previous work. We have implemented the explanation in the manuscript to make the expression more clear and precise [see Page 11, lines 5].

51. Page 16, line 2: The only positive correlation which is apparent from the table is the one of PM2.5 with dew point. The correlations with pressure and temperature are instead observed only in some situations, and are therefore probably not conclusive.

Reply: Thanks for the referee's constructive suggestions. We have seriously examined the Table 4 and rephrased the explanation to make the expression more accuracy [see

Page 11, lines 6-10].

52. Page 16, lines 2-6: The correlation of PM2.5 with relative humidity can be explained by some physical mechanisms. Please try to explain.

Reply: Thanks for the referee's suggestions. The hygroscopocity and coagulation of particulate matter has been used to explain the particle formation mechanism in the updated revision [see Page 11, lines 6-10].

53. Page 16, lines 8-22: Conclusions should be greatly revised both for the language as well as after the appropriate revisions concerning other sections have been under-taken. Reply: Thanks for the referee's suggestions. Conclusions section has been rephrased and the corresponding revisions have been implemented in new manuscript [see Page 13, lines 3, Page 14, lines 19].

Technical comments:

54. age 1, line 13: Delete "for". Change "in one" with "a".

Reply: Thanks. In the new manuscript, we have deleted "for" and changed "in one" with "a" according to the referee's suggestion [see Page 1, lines 13].

55. Page 1, line 14: Delete "the".

Reply: Thanks for the referee's comment. In the revision, we have deleted "the" [see Page 1, lines 14].

56. Page 1, line 16: Change "during the data collection" with "along the flight".

Reply: Thanks for the referee's comment. In the revision, we have changed "during the data collection" with "along the flight" [see Page 1, lines 16].

57. Page 1, line 18: Change "bigger" with "larger".

Reply: Thanks. This sentence has been deleted according to the revised conclusion [see Page 1, lines 18].

[Figure]

58. Page 1, line 21: Change "in" with "on", delete "on".

Reply: Thanks for the referee's suggestions. In the revision, we have change "in" with "on" and deleted "on" [see Page 1, lines 18].

59. Page 1, line 28: Change "of significance" to "significant".

Reply: Thanks for the referee's suggestion. We have change "of significance" to "significant" [see Page 2, lines 2].

60. Page 1, line 29: Change the second "on" to "at".

Reply: Thanks. We have changed the second "on" to "at" [see Page 2, lines 4].

61. Page 2, line 1: Delete "The only". Change "were" to "are". Change "further research" to "a correct understanding".

Reply: Thanks for the referee's suggestions. We have deleted "The only", changed "were" to "are" and changed "further research" to "a correct understanding", separately [see Page 2, lines 5].

62. Page 2, line 3: Change "model" to "models".

Reply: Thanks. We have changed "model" to "models" [see Page 2, lines 7].

63. Page 2, line 5: Delete "the". Change "concentrations" with "profiles".

Reply: Thanks for the comment. According to the referee's suggestions, we have deleted "the" and changed "concentrations" with "profiles" [see Page 2, lines 11].

64. Page 2, line 8: Change "continual" with "continuous".

Reply: Thanks for the suggestion. We have deleted change "continual" with "continuous" in the revision [see Page 2, lines 12].

65. Page 2, line 9: Change "to monitoring" to "in terms".

Reply: Thanks. We have changed the second "on" to "at" [see Page 2, lines 12].

66. Page 2, line 18: Change "efficiency" to "efficient".

Reply: Thanks for the referee's suggestions. We have changed "efficiency" to "efficient" [see Page 2, lines 16].

67. Page 2, line 21: "responsible" is not the correct term here.

Reply: Thank the referee for pointing out this question. It has been revised as "Manned aerial vehicles can undertake. . ." [see Page 2, lines 17].

68. Page 2, line 30: Change "Is of good quantitative performance" with "has a good performance".

Reply: Thanks for the comment. It has been revised based on the referee's suggestioni [see Page 3, lines 6].

69. Page 3, line 4: Change "a" with "the".

Reply: Thanks. We have changed "a" with "the" [see Page 3, lines 7].

70. Page 3, lines 6-7: Rephrase: "A total of 20 monitoring flights over half a year were carried out over the suburban area of Lin.an in YRD."

Reply: Thank you. It has been revised according to the referee's suggestion [see Page3, lines 8-10].

71. Page 3, lines 8-9: Rephrase: "The diurnal variation of PM2.5 as well as its ac-cumulation and dissipation in the atmosphere were captured and are discussed in the present article."

Reply: Thanks for the comment. The sentence has been rephrased according to the referee's suggestions [see Page 3, lines 10-11].

72. Page 3, line 10: Delete the second "the".

Reply: Thanks for the referee's suggestions. We have deleted the second "the" [see Page 3, lines 11].

73. Page 3, line 13: Change "were" to "was".

Reply: Thank you. It has been revised as "Mobile 3D measurements were…" [see Page 3, lines 15].

74. Page 3, line 13: Change "were" to "was".

Reply: Thanks for the comment. We have corrected the mistake [see Page 3, lines 15].

75. Page 3, line 14: Add "distant" before "from".

Reply: Thanks for the referee's suggestions. We have added "distant" before "from" [see Page 3, lines 18].

76. Page 3, line 16: Change "near around" to "close".

Reply: Thanks. We have changed "near around" to "close" [see Page 3, lines 21].

77. Page 4, line 5: Change "its" to "a" and "lands" to "landed".

Reply: Thank you very much. We have changed "its" to "a" and "lands" to "landed" [see Page 4, lines 12].

78. Page 4, line 6: Add a space before "300".

Reply: Thanks for the comment. We have corrected the mistake [see Page 4, lines 12].

79. Page 4, line 6: Add "of about" before "120" and delete $\sim$.

Reply: Thanks for the referee's suggestion. We have added "of about" before "120" and delete $\sim$ [see Page 4, lines 12].

80. Page 5, line 5: Change "on a total" with "for".

Reply: Thanks for the suggestion. We have changed "on a total" with "for" [see Page4, lines 15].

81. Page 5, line 6: Delete "including 16 flights". Rephrase the second sentence: "Four

flights . . ., for a total of 16 flights (Table 2)."

Reply: Thanks for the referee's suggestions. We have deleted "including 16 flights" and rephrased the second sentence [see Page 6, lines 3].

82. Page 6, line 2: Move "such as remaining battery and storage space" before "were".

Reply: Thanks. The sentence has been rephrased according to the referee's suggestion [see Page 7, lines 4].

83. Page 6, lines 2-3: Rephrase: ". . . and a visual inspection was conducted to determine the eventual compression of inlet tubing at curve."

Reply: Thanks for the referee's suggestion. We have rewritten the sentence based on the referee's suggestion [see Page 7, lines 5].

84. Page 6, line 4: Delete "allowed to" and change the tense of "warm up".

Reply: Thanks for the referee's suggestions. It has been corrected [see Page 7, lines 6].

85. Page 6, line 11: Rephrase: ". . .), maintained by the Meteorological Bureau of Lin'an."

Reply: Thanks for the referee's suggestions. The sentence has been rephrased according to the referee's suggestion [see Page 7, lines 6].

86. Page 6, lines 11-16: Rephrase: "Sounding meteorological data (air temperature, dew point temperature, relative humidity, wind speed and wind direction) from the sounding station located in Hangzhou, China, located about 40 km away far away from the experimental site which operates soundings at 12:00 UTC every day were downloaded from the University of Wyoming (. . .)."

Reply: Thanks. The sentence has been rephrased according to the referee's suggestion [see Page 6, lines 7-9].

[Figure]

87. Page 6, line 22: Change "when the UAV was taking off" to "during take-off".

Reply: Thanks for the referee's advice. We have changed "when the UAV was taking off" to "during take-off" [see Page 7, lines 13].

88. Page 8, Figure 3: The Figure has a bad resolution. Moreover, scale units for the legend should be provided.

Reply: Thanks for the referee's advice. We have re-plotted the figure with the high resolution and add scale units for the legend [see Fig. 3 on Page 8].

89. Page 9, line 2: "fligh-3" should be "flight-3". Delete the second comma. Add "as" before "afternoon".

Reply: Thanks for the referee's suggestions. The mistake has been corrected. The second comma has been deleted. "As" has been added before "afternoon" [see Page 9, lines 2].

90. Page 9, line 3: Move "increasing" before "altitude".

Reply: Thanks for the referee's comment. It has been revised [see Page 9, lines 3].

91. Page 9, line 4: Change "depicts" to "correspond to". Unify the two sentences as: "..., 2013) consistently with results from tower observations (...)".

Reply: Thanks for the referee's suggestions. We has changes "depicts" to "correspond to" and rephrased the sentences in the updated manuscript [see Page 9, lines 3].

92. Page 9, line 7: Delete "in the".

Reply: Thanks for the referee's suggestions. It has been revised [see Page 9, lines 7].

93. Page 9, line 13: Change "more" to "higher".

Reply: Thanks for the referee's suggestions. "More" has been changed to "higher" [see Page 9, lines 12].

94. Page 9, line 16: Change "Ding (Ding et al., 2005)" to "Ding et al. (2005)".

Reply: Thanks for the referee's advice. It has been revised in the updated manuscript [see Page 9, lines 14].

95. Page 13, line 4: Change "particle" to "particulate".

Reply: Thanks for pointing out the mistake. It has been corrected [see Page 14, lines 6].

Reply to the Second Referee – AMT-2016-57

Reviewer #2:

General comments:

My opinion is based on that as far as measurement methods, data processing, or algorithms or such are concerned, there are actually no new technical or methodological innovations or observations in this paper. Numerous aerosol measurements using UAVs have been published during the last few decades. So, as a concept there is nothing new. The instruments installed in the aircraft are all commercial, made by well known manufacturers. In the data processing there are essentially no new innovations either. However, as I wrote above, the results of this work are useful, for instance for evaluating the performance of air pollution forecasting models. Therefore a more suitable journal would be one that has the atmosphere itself as its focus, not methods as in AMT.

Reply: Thanks for the referee's comments and suggestions. The UAV sampling platform presented in the manuscript is proved to be of utility and feasibility, since it utilizes commercial monitors. This means this UAV sampling platform is really a "platform", because researchers can apply the UAV sampling platform to different study, just changing the onboard monitors easily.

What's more, most of the statements have been rewritten or revised based on the

referee's constructive suggestions. The passages which were ambiguous or difficult to read in our original manuscript have been rewritten. The technique details, especially the innovative part, are emphasized and expressed in the manuscript. We hope our study can be clearly presented in this revised manuscript.

Specific comments:

96. P1,L28: I would recommend changing the word "dissipation" to dispersion. Actually, the word "dissipation" is used in several sentences of the paper but in none of them it is really the good term.

Reply: Thanks for the referee's suggestion. The word "dissipation" has been changed to "dispersion" in the revised manuscript [see P1, L2; P3, L6; P3, L11; P15, L1; P15, L6; P15, 14, P16, L5; P16, L18].

97. P2,L1: " The only ground-based observations were not sufficient..." Rewrite as "Ground-based observations only are not sufficient..."

Reply: Thanks for the referee's suggestion. The sentence has been rewritten in the revision [see Page 2, Lines 5].

98. P2,L5-6 : " Studies ... were routinely conducted by meteorological tower..." This means that there was a meteorological tower who conducted studies. In other words, the expression "were conducted by" gives an impression of that the tower is a person. And the tense is misleading: "were conducted" suggests that they are not conducted any more.

Reply: Thanks to the referee for pointing out the mistakes. These tense errors have been corrected. It has been modified elsewhere throughout the manuscript as well [see Page 2, Lines 10].

99. P2,L8-9: " .. it is limited to monitoring elevation (no more than 350 m) and mobility." This would mean something is monitoring elevation, in other words monitoring whether something is rising or falling. And "monitoring mobility" would mean monitoring, whether something is moving. Rewrite.

Reply: Thanks. These sentences have been rewritten according to the referee's suggestion [see Page 2, Lines 12-13].

100. A related note: the word "monitoring" is used throughout the text in an uncorrect way. In aerosol science monitoring generally means long-lasting, continuous measurements. For example at an air quality measurement station. The UAV measurements in this paper are not monitoring, unless the flights are more or less continuous, which they are not.

Reply: Thanks to the referee's suggestions. We fully agree with the referee's comments and the corresponding section has been rewritten for more precise and serious expression [see Page 2, Lines 17, Page 3, Lines 12, 15, et al.,].

101. P2, L25: " unmanned aircraft vehicle" Should be "unmanned aerial vehicle"

Reply: Thanks for pointing out the mistake. We has corrected it in the revision [see Page 2, Lines 10].

102. Section 2.2 What is the manufacturer and model of the aircraft? What is its fuel.

Reply: Thanks for the referee's comments. This fixed-wing UAV was modified from an air-mapping aircraft manufactured by the Second Surveying and Mapping Institute of Zhejiang Province, but model of the aircraft hasn't been designated. The fuel of the engine is gasoline. Additionally, the fuel tank capacity and the engine power have been added in the new manuscript [see Page 4, Lines 6-11].

103. Table 1. The aethalometer does not measure pressure. What did you use for measuring p?

Reply: Thank the referee for pointing out the mistake. In our experiments, BC sensor (Aethlabs AE51) and Ozone sensor (POM Ozone Monitor, 2B Technologies, Inc.) were on board besides PM sensor. The pressure was actually measured by the pressure

sensor built-in POM. However, the type of the pressure sensor is not provided in the operation manual, and we have consulted the manufacturer by email but haven't received the response. The mistake has been corrected in the revision [see Table 1 on Page 5].

104. P5.L2-4: Describe the results of the intercomparison better than giving just one correlation coefficient. How long was the intercomparison, show scatter plots, and regression lines, slope and offset.

Reply: Thanks for the referee's suggestion. The comparison experiment lasted for 21 days and was conducted under different temperature and humidity conditions. The AM510 data is quite consistent with the TEOM data, with correlation coefficient (R) of 0.99, intercept of -11.52 and slope of 0.82 [see Page 5, Lines 9-11]. The picture blow shows the comparison between AM510 and TEOM data on 4 days.

105. Table 2 needs a proper, detailed caption and column explanations.

Reply: Thanks to the referee's suggestion. The caption of Table 2 has been refined and brief explanations of Table 2 have been implemented into new manuscript according to the referee's advice [see Page 5, Lines 2-5].

106. P6,L21 What does "self-monitoring" mean?

Reply: Sorry for the confusion. What we want to express in the manuscript is that the sampling PM data, could be contaminated by the exhaust from the UAV engine. We have rephrased the sentences in the revision to avoid misunderstanding [see Page 6, Lines 11-12].

107. P7,L5. The formula (1) is not from Day et al. (2000), where is it from? They gave growth of scattering coefficient and it is not the same as the growth of mass. And it varies with the chemical composition. –what is the reasoning for using this CF? And it is unclear from the explanation, how did you use the CF. Did you correct with it all PM concentrations to RH 0% or what? Secondly, if I put in the formula any RH > 2 and

100, the formula gives negative values. Correct it.

Reply: Thanks for the referee's comment. The formula is referred to the note (Dusttrak DRX Aerosol Monitor in Environmental Application Note EXPMN-066). We fully agree with the referee's opinion that the scattering coefficient varies with the chemical composition. To calibrate the sampled PM data, the data were cleaned as mentioned in the updated manuscript [see Page 7, Lines 22]. Then the data were calibrated with the formation to eliminate the influence of humidity. Considering the difference in chemical compositions, the correction coefficient was obtained by the comparison test between the AM 510 with TEOM instruments at Shanghai [see Page 5, Lines 7-11]. As for the negative values, the RH value ranges from 0% to 100%, therefore the calculated value remains positive. Sorry for the confusion. And more technical explanation are provided in the updated manuscript.

108. In the time series figures, add dates in the x-axes to make it easier to read.

Reply: Thanks for the referee's suggestions. We have re-plotted these time series figures and have add dates in the x-axes [see Fig. 6 on Page 13, Fig. 8 on Page 14, Fig. 9 on Page 15].

109. Move section 3.4 earlier because the met data are used in the explanations of the profiles.

Reply: Thanks for the referee's suggestions. We has reconstructed the sections in the revision [see Page 10, Lines 3 - Page 13, Lines 4].

Please also note the supplement to this comment:
http://www.atmos-meas-tech-discuss.net/amt-2016-57/amt-2016-57-AC1-supplement.zip
* * *
[Figure]

[Figure]

**Fig. 1.** Left:UAV and instruments arrangement; Right:the flight route and 3D terrain landscape

**Fig. 2.** Spatial distribution of PM2.5 concentrations

[Figure]

**Fig. 3.** Surface PM2.5 concentrations, relative humidity, wind speed and wind direction measured by Lin'an regional background station (30°18' N, 119°44' E) on 12th-16th November

**Fig. 4.** Surface PM2.5 concentration, relative humidity, wind speed and wind direction measured by Lin'an regional background station (30°18' N, 119°44' E) on 10th -14th December

**Fig. 5.** Surface PM2.5 concentrations, humidity, wind speed and wind direction measured by Lin'an regional background station (30°18' N, 119°44' E) on 3rd-7th February

- Y-axis: PM2.5 measured by TEOM mass measurements (ug/m^3), ranging from 0 to 500
- X-axis: PM2.5 measured by SidePak (ug/m^3), ranging from 0 to 500

$$y = 0.8334x - 10.441$$
$$R^2 = 0.9929$$

**Fig. 6.** comparison between AM510 and TEOM data on four days